# Preparation and Properties of Chitosan/Graphene Modified Bamboo Fiber Fabrics

**DOI:** 10.3390/polym11101540

**Published:** 2019-09-21

**Authors:** Yan Wu, Yuqing Bian, Feng Yang, Yang Ding, Kexin Chen

**Affiliations:** 1College of Furnishings and Industrial Design, Nanjing Forestry University, Nanjing 210037, China; lndr810@163.com (Y.B.); d850297003@outlook.com (Y.D.); chencanx@outlook.com (K.C.); 2Co-Innovation Center of Efficient Processing and Utilization of Forest Resources, Nanjing Forestry University, Nanjing 210037, China; 3Fashion Accessory Art and Engineering College, Beijing Institute of Fashion Technology, Beijing 100029, China

**Keywords:** chitosan, graphene, fabric, antimicrobial properties

## Abstract

Chitosan (CS) and graphene (Gr) were used to modify bamboo fiber fabrics to develop new bamboo fiber fabrics (CGBFs) with antimicrobial properties. The CGBFs were prepared by chemical crosslinking with CS as binder assistant and Gr as functional finishing agent. The method of firmly attaching the CS/Gr to bamboo fiber fabrics was explored. On the basis of the constant amount of CS, the best impregnation modification scheme was determined by changing the amount of Gr and evaluating the properties of the CS/Gr modified bamboo fiber fabrics. The results showed that the antibacterial rate of CGBFs with 0.3 wt% Gr was more than 99%, and compared with the control sample, the maximum tensile strength of CGBF increased by 1% in the longitudinal direction and 7.8% in the weft direction. The elongation at break increased by 2.2% in longitude and 57.3% in latitude. After 20 times of washing with WOB (without optical brightener) detergent solution, the antimicrobial rate can still be more than 70%. Therefore, these newly CS/Gr modified bamboo fiber fabrics hold great promise for antibacterial application in home decoration and clothing textiles.

## 1. Introduction

Chitin deacetylates to form chitosan, which has good biocompatibility, degradability, bactericidal and bacteriostatic effects in different environments [1]. Graphene, as a two-dimensional carbon nano-material, has attracted much attention due to its excellent mechanical, thermal, optical and electrical properties. Graphene and its composites can also form hydrogen bonds with biological molecules on the cell wall through oxygen-containing groups such as carboxyl and hydroxyl groups on the graphene lamellae, which can isolate the cytoplasm of bacteria and eventually cause the bacteria to lose nutrients and die [2]. Zhao et al. [3] prepared graphene oxide-based antibacterial cotton fabric by direct adsorption, radiation crosslinking and chemical crosslinking. Abate et al. [4] modified polyester fabric with chitosan to optimize its antimicrobial and hydrophobic properties. The chitosan/graphene composite film prepared by Xie et al. [5] has good mechanical and vapor transport properties.

Bamboo fibers are made of bamboo pulp and viscose fibers by wet spinning. Bamboo fibers are different from ordinary viscose fibers and have no obvious skin-core structure. Scanning electron microscopy showed that bamboo fibers had different cross-sectional sizes, uneven distribution of microporous structure, and a large number of grooves and hollow grooves, which made bamboo fibers have good hygroscopicity and air permeability, comfortable handle [6], and provided suitable matrix for microbial growth. Song et al. [7] introduced chitosan into bamboo fabric by surface modification process, which improved the antimicrobial property of textiles.

In related studies, the chitosan/graphene composites were used to prepare hydrophilic fabrics by adding silane coupling agent by layer-by-layer self-assembly or direct adsorption crosslinking, which had good antibacterial properties and mechanical properties. However, the steps were relatively tedious and the washing resistance was poor. The aminopropyltriethoxysilane (APTES) crosslinking agent was innovatively used to increase the amino group of chitosan and improve its crosslinking property in this work. The operation process was simple, the cost was low, and the fabrics still had good antimicrobial activity under acid and alkali conditions after washing many times. Moreover, the relationship between fabric chroma and antimicrobial activity was found after different washing times.

In this study, the bamboo fiber fabrics were impregnated with CS/Gr composite solution to prepare for the CS/Gr modified bamboo fiber fabrics (CGBFs), and the effects of chitosan and graphene contents on the antibacterial properties of CGBFs were evaluated, respectively. The antibacterial materials (chitosan and graphene) were introduced into bamboo fiber fabrics by dipping method and a green, environmentally friendly and excellent performance fabric was developed.

## 2. Experimental Section

### 2.1. Experimental Methods

#### 2.1.1. Preparation and Characterization of CGBFs

The test temperature was controlled at about 26 °C. Six pieces of bamboo fiber fabrics with the sizes of 100 mm × 100 mm (weft direction × warp direction) were selected to flatten and remove the wool edges so as to make the surface clean and smooth. They were dried in a 60 °C oven until they were completely dried. 1 g of chitosan was dissolved in 100 mL acetic acid solution with 1% mass concentration, stirred for 30 min and then stirred for 3 h with APTES crosslinking agent. The mass fraction of APTES is 2% to the CS solution. The crosslinked chitosan solution was added with Gr of 0.1, 0.2, 0.3, 0.4 and 0.5% mass concentration, respectively. In order to express conveniently, different ratios of CS to Gr were given different sample codes as shown in Table 1.

CS/Gr mixed solution was obtained by ultrasonicating for 30 min and the sample codes were shown as: 0.1 wt% CS and 0.1 wt% Gr (C1G1); 0.1 wt% CS and 0.2 wt% Gr (C1G2); 0.1 wt% CS and 0.3 wt% Gr (C1G3); 0.1 wt% CS and 0.4 wt% Gr (C1G4); and 0.1 wt% CS and 0.5 wt% Gr (C1G5). Bamboo fiber fabrics were soaked in the CS/Gr mixed solutions for 45 min, respectively, then rinsed and dried for further tests.

#### 2.1.2. Antibacterial Rate Test

According to the national standard GB/T 20944.3-2008, the antimicrobial activity of modified bamboo fiber fabrics was qualitatively evaluated by *E. coli*. The CGBFs were tested for their antimicrobial activity. Colony count method was used to calculate the antimicrobial activity according to Formula (1). The average antimicrobial activity *R* was obtained after three counts:(1)$$R=\frac{\left(T_{0}-T_{1}\right)}{T_{0}}*100\%$$
where *T*_₀_ was the number of bacteria on the plate of blank sample and *T*_₁_ was the number of bacteria on the plate of tested sample.

#### 2.1.3. Observation of Micromorphology

In a general room temperature environment, the composite fabrics of the control sample and the modified ones were fixed on a metal bracket with conductive adhesive. Platinum plating and gold plating were sprayed after drying, and observed by environmental scanning electron microscopy (FEI Company, New York, NY, USA).

#### 2.1.4. Fourier Transform Infrared Spectrometer (FTIR) Test

The surface functional groups of the composite fabrics of the control sample and the modified ones were measured by FT-IR infrared spectrometer (Avance 300, Bruker Company, Berlin, Germany). The fabrics were completely dried and placed on the top of the carrier sheet to be flattened for infrared scanning. The transmission mode of infrared microscope was selected when testing, the frequency of spectrum scanning was 200 times, the resolution of infrared spectrum was 1.5 cm^−1^ and the range of spectrum acquisition was 4000–650 cm^−1^.

#### 2.1.5. Mechanical Properties Test

According to ISO13934-1 “Testing of Tensile Strength of Fabrics,” the fabric strength was tested by YG (B) 026D-250 strength tester (Aoran Tech. Co. Ltd., Shanghai, China). The finished fabrics were cut according to the specifications of 210 cm × 297 cm. The holding distance was 100 mm and the drawing speed was 100 mm/min. Five repeats were tested for each sample and the average values were shown in Table 2. The breaking strength and elongation at break were taken as the indexes to evaluate the fabric strength. 

#### 2.1.6. Washing Fastness Test

The washing fastness of impregnated fabrics was tested according to GB/T 12490-2014 “Textile Color Fastness Test for Family and Commercial Washing Fastness.” After washing and drying, the L, a, and b-values of each washed fabric were measured by CM-2500d color meter of Konica Minolta (Tokyo, Japan).

#### 2.1.7. Washing Resistance and Antimicrobial Test

According to the national standard GB/T 8629-2017 household washing and drying procedures for textile testing, the fabrics were washed and dried several times. According to the national standard GB/T 20944.3-2008, the antimicrobial activity of modified bamboo fiber was qualitatively evaluated by *E. coli*. The antimicrobial rate was obtained by counting the number of bacteria and taking the average value as Section 2.1.2. mentioned.

#### 2.1.8. Viscosity Test

DV-S rotary viscometer (Nanjing, China) of Nanjing Zijin Metrology Co., Ltd. was used to measure the viscosity of the impregnating solution. Pour the impregnating solution into the test container and insert the cylinder into the impregnating solution until it has completely passed the top rotor of the cylinder. Adjust the speed, turn on the motor, test and read after data stabilization. Measure three times continuously. The difference between the measured value and average value should not exceed (+3%) of the calculated average value. Otherwise, the fourth measurement should be done and finally, the average value was obtained.

## 3. Results and Discussion

### 3.1. Principle of Preparation of CS/Gr Solution

Aminos on chitosan can amide with carboxyl groups on graphene to form -NHCO- bonds, as shown in Figure 1. The amino group of chitosan is in the second place in the molecular structure and its performance is weak. Therefore, when adding APTES crosslinking agent to strengthen the amino group it is easy to form composite materials closely linked with graphene. The modification mechanism is shown in Figure 2 [8].

### 3.2. Effect of Graphene Addition on Antibacterial Property of Impregnated Fabrics

Under normal conditions, chitosan is insoluble in water, but soluble in acidic medium. In acidic medium, the amino group of chitosan molecule is protonated to form -NH_3_+, which results in acid dissolution [9]. As shown in Figure 3, the molecular structure of chitosan has amino group, and some studies have shown that amino group in chitosan molecule is the main driving force of its antimicrobial properties [10].

As shown in Figure 4, the number of *Escherichia coli* on the modified bamboo fiber fabrics dropped significantly on the plate, which means that the modified bamboo fiber fabrics have a better antimicrobial effect. The sharp edges of graphene sheets cause physical damage to bacterial cell membranes, which results in the outflow of intracellular substances and the death of bacteria [11].

The antibacterial rate of chitosan composite fabrics is shown in Figure 5. When the concentration of chitosan is 0.3%, the antibacterial rate reaches over 98% and tends to be 100%. The antibacterial rate of the chitosan composite graphene modified fabric is shown in Figure 6. When the concentration of 0.1% chitosan composite is 0.1% graphene, the antibacterial rate reaches 94%, while the antibacterial rate of 0.1% chitosan composite fabric is less than 75%. It can be seen that the addition of graphene composite fabric improves the antibacterial property of the fabric.

In addition, when the pH value of the impregnating solution is about 3.7, the impregnating solution is adjusted to the neutral one. It is found that the color of the fabric begins to yellow and the antimicrobial property decreases. This is because the antimicrobial property of chitosan is mainly based on the protonated amino group, which is in the neutral environment after neutralizing acetic acid, and the protonated amino group no longer exists. However, the fabric treated with chitosan graphene solution has a strong antimicrobial property and less environmental impact, so the addition of graphene makes up for the defect that chitosan has a poor antimicrobial property under neutral and alkaline conditions.

### 3.3. Morphology Characterization of Impregnated Fabrics with Different Graphene Additions

Figure 7 shows ESEM images of CS/Gr modified bamboo fiber fabrics (CGBFs). With the increase of graphene content, the surface adhesion of bamboo fiber fabrics increased significantly. The surface of untreated fabrics was smooth, free of impurities, and with characteristic tubular texture by ESEM. By comparison, it can be found that when the graphene content reaches 0.4 wt%, the surface morphology of treated bamboo fiber fabric is covered by graphene finishing agent, which masks the original characteristic stripes on the surface of the fiber. When the graphene content is 0.3 wt% or lower, graphene is loaded on a single fiber and will not attach between the fibers, so adding a small amount of graphene will not block the fabric, will maintain the air permeability, and improve the comfort of wearing [12]. At the same time, a lot of irregular accumulation can be found on the surface of bamboo fibers covered by finishing agents, which characterizes the graphene microsheets dispersed on the fabric. With the increase of graphene concentration, the dispersion of CS/Gr in some parts of bamboo fibers is macroscopically uneven.

### 3.4. FTIR Analysis

Figure 8 shows the infrared spectra of CS/Gr modified fabrics. It can be seen that the broad absorption peaks of 3500 cm^−1^–3300 cm^−1^ are due to the stretching of the -OH group. The characteristic absorption peaks of hydroxyl groups indicate that all samples contain cellulose. The CS/Gr spectra shows that 1364 cm^−1^ and 1017 cm^−1^ after adding graphene, correspond to C–O and C–O–C stretching correspond to graphene, respectively. The peaks at 3330 cm^−1^ and 1017 cm^−1^ are obvious, which are caused by the overlap of O–H and C–O stretching peaks of graphene with the stretching peaks of fabric cellulose. The absorption band near 1636 cm^−1^ is the in-plane stretching vibration peak of amide-NH, which indicates that the amino group in chitosan covalently binds to the carboxy group of graphene, and proves that the crosslinking of chitosan and graphene has been completed. Compared with CS/Gr composite fabrics, the stretching vibration absorption peak of -OH at 3330 cm^−1^ of crosslinked fabrics shifted to a low wavenumber (red shift) and its width narrowed, indicating that crosslinking enhanced the interaction between modifiers and fabrics [13].

### 3.5. Effect of Graphene Addition on Mechanical Properties of Impregnated Fabrics

The mechanical properties of modified fabrics change with the increase of graphene content as shown in Table 2. Graphene is a rigid material with high strength. After adding graphene, the tensile strength increases. When the amount of chitosan is 0.1%, the maximum tensile strength decreases by 4.2% in warp and 4.7% in weft. The elongation at break is unchanged in longitude and decreased by 7.8% in latitude. This shows that after chitosan finishing, the fabric becomes brittle and hard, and the tensile resilience decreases [14,15], so that the mechanical breaking strength and breaking elongation all follow. After adding graphene, the maximum tensile strength increases by 1% in longitudinal direction, 7.8% in weft direction, 2.2% in longitudinal direction of elongation at break, and 57.3% in weft direction, indicating that the mechanical properties of the fabric are improved by adding graphene.

### 3.6. Effect of Graphene Addition on Washing Resistance and Antibacterial Activity of Impregnated Modified Fabrics

Washability is an important index for evaluating fabric properties. The antibacterial rate of fabrics impregnated with 0.1% chitosan solution decreased by 32% after five washes. The reason is that acetic acid is easy to wash and remove. The fabrics with different graphene content were washed 20 times, and the antimicrobial rate was measured every 5 times and the results were recorded (Figure 9). The sterilization efficiency of these five fabrics decreased slowly with the washing times. There is a hydrogen bond between chitosan and cellulose. Graphene and cellulose fibers have poor crosslinking stability. After crosslinking of chitosan, graphene and chitosan form an enclosed structure [16]. At the same time, it was found that when graphene content was 0.3, 0.4 and 0.5 wt%, the degree of decline began to be gentle after 10 times of washing. Perhaps with the increase of graphene content, the amount of self-polymerization of cellulose long molecular chains decreased relatively, and the contact area between graphene and cellulose fibers would not be markedly reduced, thus the washing durability rate would be increased [17].

### 3.7. Color Fastness

The test chromaticity indices are L, a and b. The color characteristics of each sample are the average values of five test points. As can be seen from Figure 10a, graphene has a great influence on fabric dyeing. With the increase of graphene content, the L-value decreases gradually, which indicates that the larger the graphene content, the lower the lightness index of the fabric and the darker the color after impregnation. At the same time, it can be found that with the increase of washing times, the change of brightness index is gradually flat, which indicates that with the increase of surface adsorption, the absorbent impregnating solution is limited and some of it will be washed off, which corresponds to the results of washability and the antimicrobial test. The more graphene is added, the greater the color difference before and after washing. The lighter tones of deeper tones have lower firmness when washed and wet rubbed. Because the dye molecules are more saturated in deeper tones, it is easier to remove from the inside of the fibers during washing [18].

Figure 10b shows that with the increase of graphene content, the red–green chromaticity index a-value gradually decreases and maintains a positive number, but with the increase of washing times, the a-value shows a downtrend, indicating that the fabric has been greened after washing. At the same time, as shown in Figure 10c, the yellow–blue chromaticity index b-value gradually increases, indicating that the fabric blues after washing, which makes the color of the fabric lighter [19].

### 3.8. Effect of Graphene Addition on Viscosity of Impregnating Solution

Under different control variables, the results of the viscosity test are shown in Figure 11. With the increase of graphene content, the viscosity of the impregnating solution fluctuates slightly, but the overall trend is gradually rising. It shows that graphene can increase the viscosity of the impregnating liquid system when other conditions remain unchanged. The size distribution of graphene particles used in this experiment is wide. All particles are very small and most of the particles reach the micron level [8]. This shows that although the size of graphene powder after grinding decreases, there is still agglomeration phenomenon. At the same time, the chitosan solution itself has a certain viscosity. The viscosity of the impregnating solution increases slightly after adding graphene. The reason may be that with the increase of graphene mass fraction, the distance between graphene lamellae decreases, the probability of contact increases, and agglomeration easily occurs, which leads to the increase of graphene sheet diameter. The viscosity of the impregnating solution increases with the increase of graphene mass fraction [20].

## 4. Conclusions

Bamboo fiber fabrics were impregnated with chitosan/graphene solution, which gave the obtained fabrics excellent antimicrobial and mechanical properties. When the mass ratio of graphene was 0.2%, the antimicrobial rate of the fabrics reached more than 98%. ESEM images showed that the impregnating modifier (CS/Gr) had been grafted onto the fabric. When the mass ratio of graphene was below 0.3%, no bridge was erected between the fibers, which ensured good air permeability of the fabric. FTIR characterization also confirmed that the amino group of chitosan combined with the carboxyl group of graphene to produce the amide group, proved that the CS/Gr and fabric were successfully crosslinked. The addition of chitosan made the fabric brittle and the mechanical properties weakened. The tensile strength and elongation at break of the fabric increased after adding graphene, and the mechanical properties of the fabric increased. The antibacterial rate of the fabrics with the composite mass ratio of 0.1 wt% graphene remains above 70% after 20 washes, while that of the fabrics treated with 0.1% chitosan after 5 washes decreases by 32% due to the lack of strong bonding between chitosan and fabrics, and the removal of acetic acid by washing. At the same time, the color of the modified fabric is less affected by repeated washing and after washing, the fabric will be greened and blued. To sum up, CS/Gr modified fabric is a kind of low cost, high efficiency, green and environmental protection textile.

## Figures and Tables

**Figure 1 polymers-11-01540-f001:**
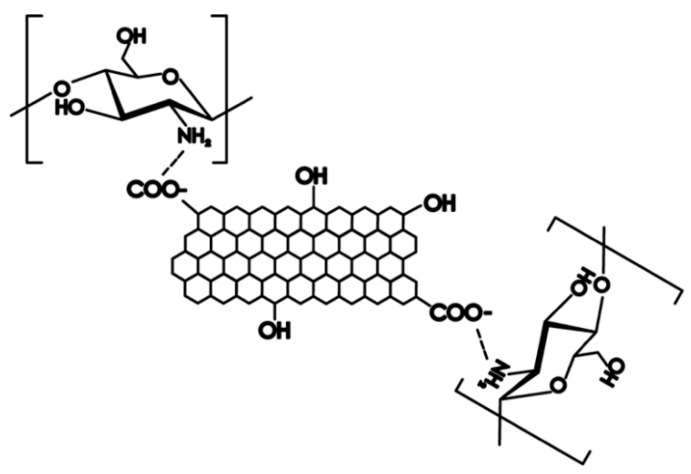
Molecular schematic diagram of amidation reaction.

**Figure 2 polymers-11-01540-f002:**
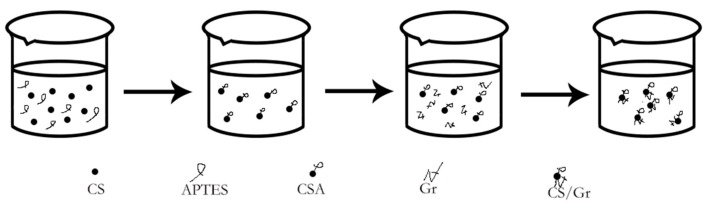
Modification mechanism of CS/Gr modified bamboo fiber fabrics (CGBFs).

**Figure 3 polymers-11-01540-f003:**
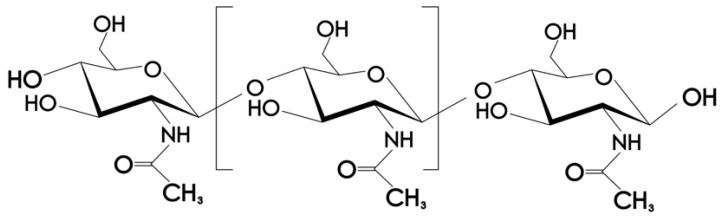
Molecular formula of chitosan.

**Figure 4 polymers-11-01540-f004:**
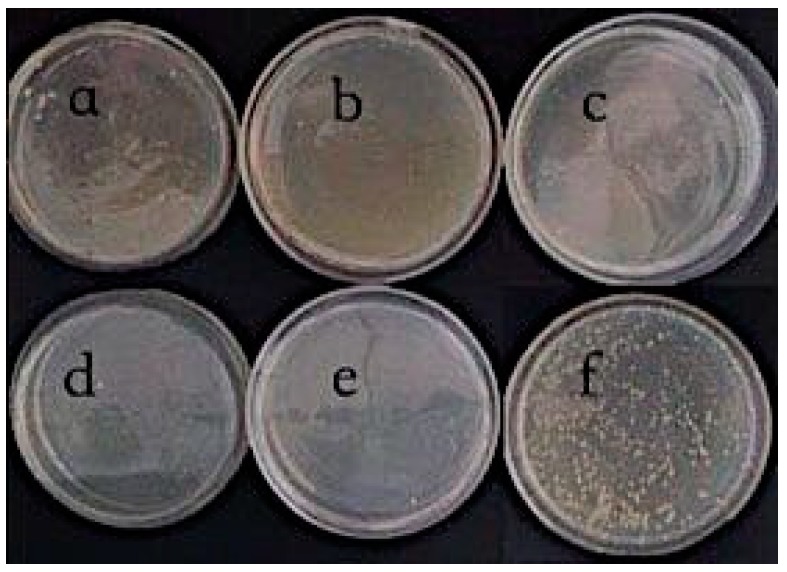
Photographs of antibacterial activity of CS/Gr modified bamboo fiber fabrics (CGBFs). (**a**) C1G1; (**b**) C1G2; (**c**) C1G3; (**d**) C1G4; (**e**) C1G5; and (**f**) control sample.

**Figure 5 polymers-11-01540-f005:**
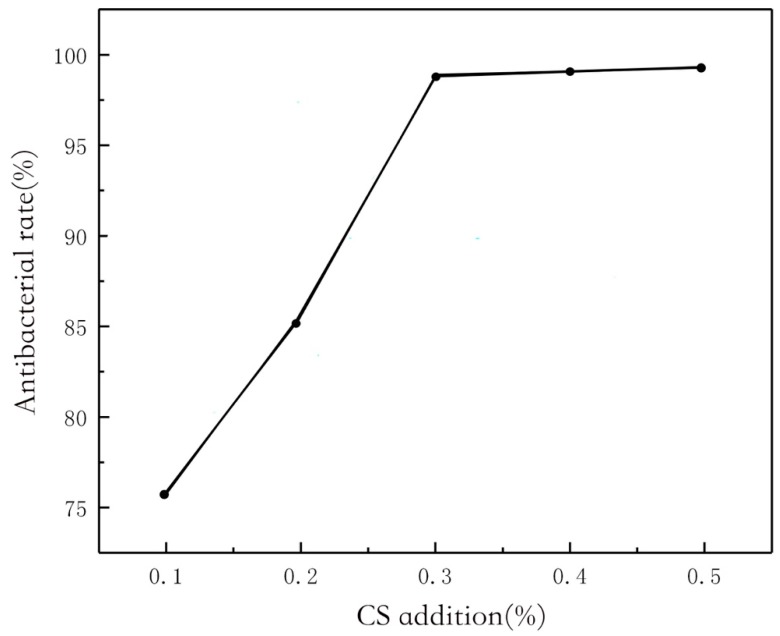
Effect of CS addition on the antimicrobial activity of CS/Gr modified bamboo fiber fabrics (CGBFs).

**Figure 6 polymers-11-01540-f006:**
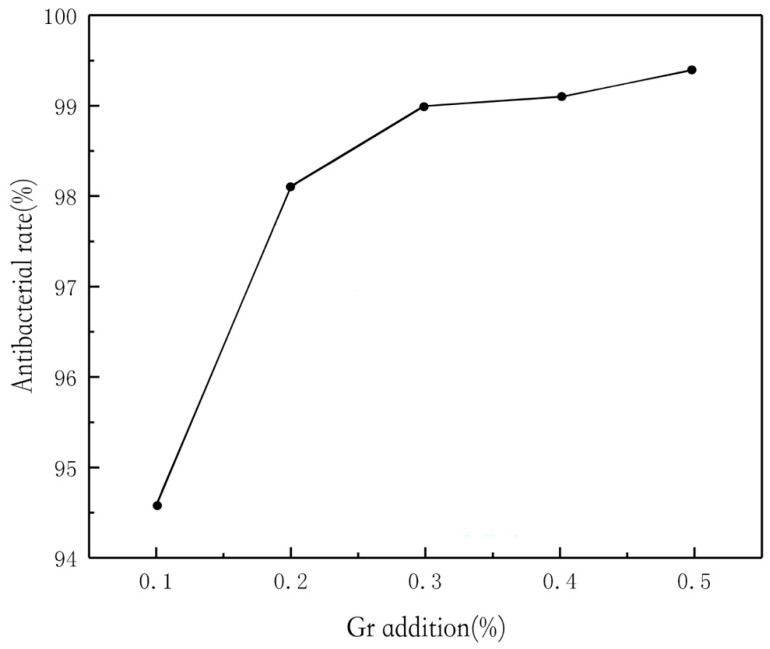
Effect of Gr addition on the antimicrobial activity of CS/Gr modified bamboo fiber fabrics (CGBFs).

**Figure 7 polymers-11-01540-f007:**
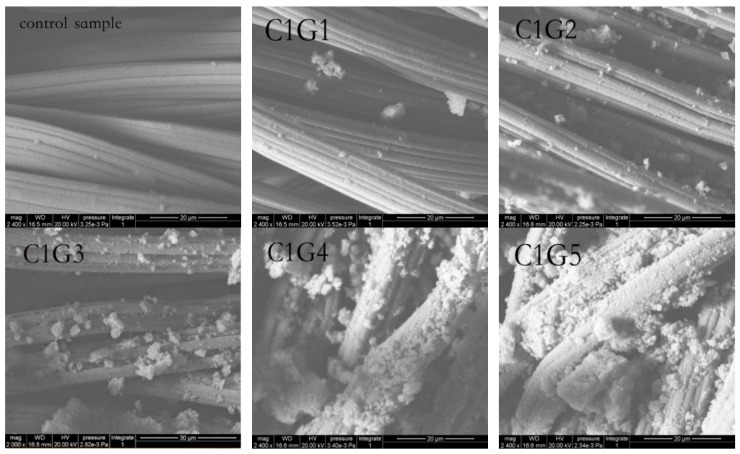
ESEM images of CS/Gr modified bamboo fiber fabrics (CGBFs).

**Figure 8 polymers-11-01540-f008:**
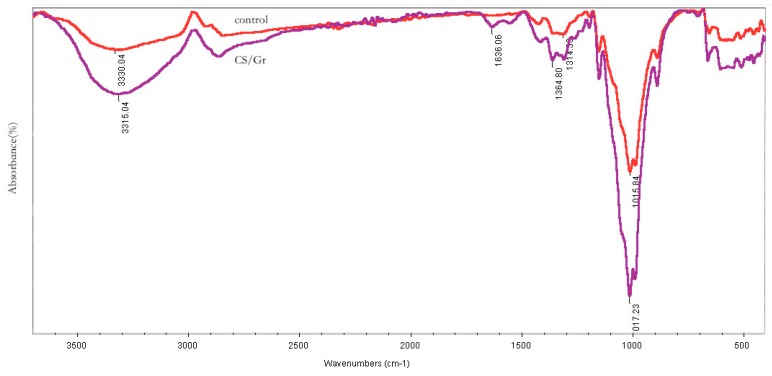
Infrared spectra of control and CS/Gr modified fabrics.

**Figure 9 polymers-11-01540-f009:**
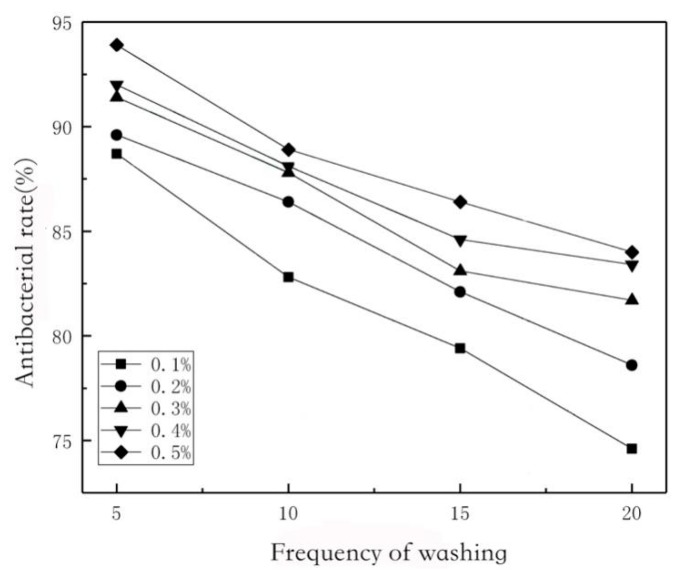
Washing resistance antibacterial rate of CS/Gr modified bamboo fiber fabrics (CGBFs) with different graphene additions.

**Figure 10 polymers-11-01540-f010:**
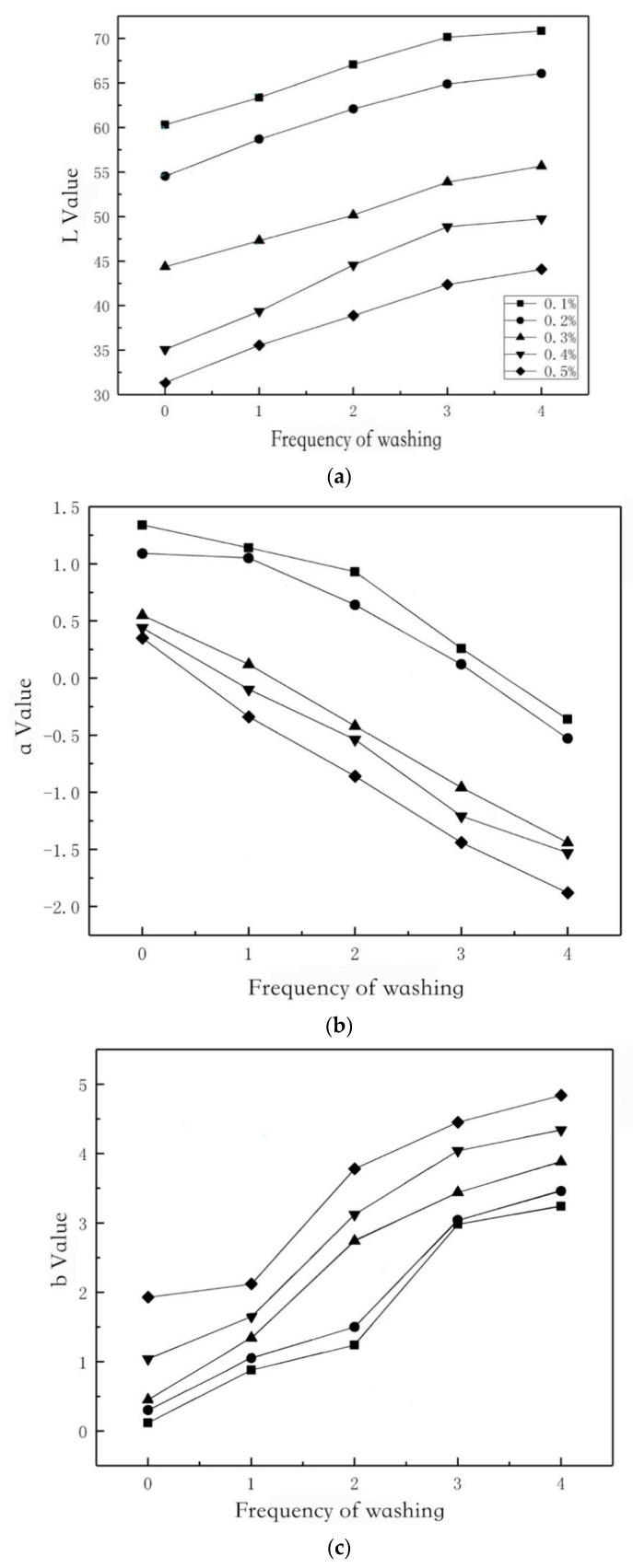
The relation between chromaticity indices (L*a*b) and washing number of CS/Gr modified bamboo fiber fabrics (CGBFs) with different graphene addition. (**a**) L-value; (**b**) a-value; and (**c**) b- value.

**Figure 11 polymers-11-01540-f011:**
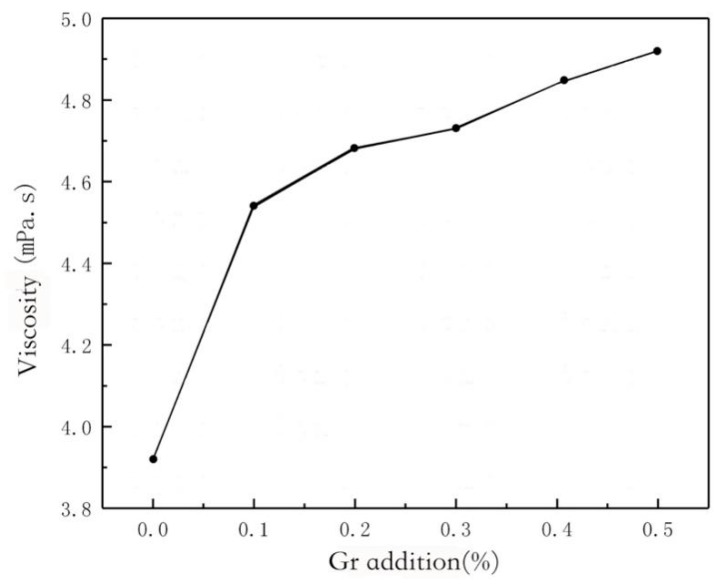
Relationship between viscosity and Gr addition.

**Table 1 polymers-11-01540-t001:** Sample codes of different ratios of chitosan (CS) to graphene (Gr).

CS (%)	Gr (%)	Sample Code
0.1	0.1	C1G1
0.1	0.2	C1G2
0.1	0.3	C1G3
0.1	0.4	C1G4
0.1	0.5	C1G5

**Table 2 polymers-11-01540-t002:** Mechanical properties of control sample, CS and CS/Gr modified fabrics.

	Maximum Tensile Strength (Mpa)		Elongation at Break (%)	
	*X* _1_	*Y* _1_	*X* _2_	*Y* _2_
**control**	204.6 (3.6)	227 (3.5)	22.3 (0.2)	30.7 (0.3)
**C1**	196 (3.3)	216.3 (3.0)	22.3 (0.2)	28.3 (0.4)
**C1G3**	205.8 (3.5)	244.6 (3.2)	22.8 (0.3)	47.2 (0.2)

*X_1_* is the longitudinal maximum tensile strength, *Y_1_* is the latitudinal maximum tensile strength, *X_2_* is the longitudinal breaking elongation, and *Y_2_* is the latitudinal breaking elongation. The data in the parent is the standard deviation (SD).

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
