# Peer review of "Preparation and Properties of Chitosan/Graphene Modified Bamboo Fiber Fabrics"

_polymers, 2019, doi:10.3390/polym11101540_

Round 1
Reviewer 1 Report
The work by Wu et al. deals with a very interesting topic: the modification of a natural fiber with a mixture of chitosan and graphene, which can present interest for biomedical industry with the aim to fabricate antibacterial materials. However, the authors, does not take care on the presentation of the work and they present a merely descriptive work without a real scientific discussion. Some specific points:
-The abstract should be rewritten in a more concise way to ensure that the main insights of the work are properly summarized
-Introduction seems to be a set of sentences without any apparent connection.
-The test equipment and instrumentation section is not needed with experimental methods one is enough.
-There are no discussion of the main findings, authors only present a description of it.
-The english should be revised within the manuscript.
Author Response
Dear Editor and Reviewers:
I am pleased to resubmit for the revised version of manuscript entitled “Preparation and properties of chitosan/graphene modified bamboo fiber fabrics”. Thank you for reading our manuscript and reviewing it. Those comments are all valuable and very helpful for revising and improving our paper. We have revised our manuscript carefully and have made correction which we hope meet with approval. So we have sent the revised manuscript and have highlighted changes by using the track change mode. The main corrections in the paper and the responds to the reviewers’ comments are as following:
Response to the reviewer 1:
The abstract should be rewritten in a more concise way to ensure that the main insights of the work are properly summarized.
Answer: The abstract was rewritten and have deleted some descriptive sentences to highlight the application of modified fabrics. The new abstract is as follows: “Chitosan (CS) and graphene (Gr) were used to modify bamboo fiber fabrics to develop new bamboo fiber fabrics (CGBFs) with antimicrobial properties. The CGBFs were prepared by chemical crosslinking with CS as binder assistant and Gr as functional finishing agent. The method of firmly attaching the CS/Gr to bamboo fiber fabrics was explored. On the basis of the constant amount of CS, the best impregnation modification scheme was determined by changing the amount of Gr and evaluated the properties of the CS/Gr modified bamboo fiber fabrics. The results showed that the antibacterial rate of CGBFs with 0.3 wt% Gr was more than 99 %, and comparing with the control sample, the maximum tensile strength of CGBF increased by 1 % in the longitudinal direction, 7.8 % in the weft direction, the elongation at break increased by 2.2 % in longitude and 57.3 % in latitude. After 20 times of washing with WOB detergent solution, the antimicrobial rate can still be more than 70 %. Therefore, these newly CS/Gr modified bamboo fiber fabrics hold great promise for antibacterial application in home decoration and clothing textiles.”
Introduction seems to be a set of sentences without any apparent connection.
Answer: In the introduction, the adhesives, modifiers and carriers are explained. Finally, the preparation and application of composite fabrics are described.
The test equipment and instrumentation section is not needed with experimental methods one is enough.
Answer: Ok, the test equipment and instrumentation section have been deleted .
There are no discussion of the main findings, authors only present a description of it.
Answer: In the results and discussion part, the effects of graphene addition on antibacterial property, washing resistance and antibacterial activity, morpology characterization, color fastness, viscosity of CS/Gr solution of impregnated fabrics. The CS combined with Gr could improve the antibacterial property, washing resistance of fabrics. After 20 times of washing with WOB detergent solution, the antimicrobial rate can still be more than 70 %. Aminos on chitosan can amide with carboxyl groups on graphene to form - NHCO - bonds, according expriment and analysi, we also found that adding APTES crosslinking agent to strengthen amino group is easy to form composite materials closely linked with graphene. From the FTIR analysis, the absorption band near 1636 cm-1 is the in-plane stretching vibration peak of amide-NH, which indicates that the amino group in chitosan covalently binds to the carboxy group of graphene, and proves that the crosslinking of chitosan and graphene has been completed. In the future work, we will further study the modification processing, improving the adhesion of Gr on the fabrics and antimocrobial rate after more times washing.
The English should be revised within the manuscript.
Answer: The English language was revised carefully.
We appreciate for Editor and Reviewers’ warm work earnestly, and hope that the correction will meet with approval. Once again, thank you very much for your comments and suggestions.
Yours sincerely,
Yan Wu and Bian Yuqing

Reviewer 2 Report
Is 70% antimicrobial rate acceptable. Also, the final application field is not described. Materials and reagents are listed rather than describing their significance in materials section A list of equipments and instruments are listed as in a table. No test conditions or processing details are described. Perhaps, these sections can be improved. What is the moisture regain of bamboo fibers and what do authors mean by completely dried? Bone-dry? How much amount of APTES was added? What is the significance of choosing and using APTES cross-linking agent? Table 1 has listed CS% as 0.1% wheres text from 111-115 is describing as 1% of CS. Inconsistent use. What test conditions were followed for section 1.3.3 At what resolution were IR spectra capture? Author listed that graphene sheets cause physical damage, is this validated in current work and any data? How statistically significant is data reported in section 2.2? Matured cotton's longitudinal cross-sectional images are normally collapsed tube and kidney like structures and not striped textured. Needs confirmation and correction of reporting either cotton or bamboo fibers Why are authors referring to reference 13 when observations are evident from Figure 7. Is Figure 7 from previous work? If so, needs to clearly differentiate. Table 2 is just referring to control, C1 and C1G3, how is the trend looking for other loading ratios? And why are authors limiting sharing data for just these and not whole dataset? Anti-microbial rate after 10 washes is reported as gradual and gentle decrease. But from 90% to 75% drop is significant. Is this decrease acceptable? And how reproducible is this data? Relative decrease after 10 washes is reported as self polymerization and contributing to increase in wash durability. This is reported as a theory and needs further validation in reference to current work. Also, Figure 9 is demonstrating decreasing trend whereas text is reported as increasing. Contradicting statements and observations. Needs further clarification. Needs clarification of representing L, a, b and/or L*, a*b as reported in Figure 10 What does authors mean by agglomeration phenomenon and how does it apply in the current work? Only tensile properties are reported. How do other textile properties vary? Burst strength, air permeability, weight pick-up, bending rigidity, drape? These are not reported Through out the manuscript, C1 has 75%, C3 has 100% of antimicrobial rates, and G1 = 94.5%, G3 = 98%. Based on these observations, should C3G3 be the best combination? Why is final loading restricted to C1G3? Conclusion section needs work on describing and summarizing the key highlights.
Author Response
Dear Editor and Reviewers:
I am pleased to resubmit for the revised version of manuscript entitled “Preparation and properties of chitosan/graphene modified bamboo fiber fabrics”. Thank you for reading our manuscript and reviewing it. Those comments are all valuable and very helpful for revising and improving our paper. We have revised our manuscript carefully and have made correction which we hope meet with approval. So we have sent the revised manuscript and have highlighted changes by using the track change mode. The main corrections in the paper and the responds to the reviewers’ comments are as following:
Response to the reviewer 2:
Is 70% antimicrobial rate acceptable?
Answer: According to the standard of GB/T 20944-2008, the evaluation of the antimicrobial effect of Escherichia coli is greater than or equal to 70 %, which indicates that the sample has antimicrobial activity.
Also, the final application field is not described. Materials and reagents are listed rather than describing their significance in materials section A list of equipments and instruments are listed as in a table. No test conditions or processing details are described. Perhaps, these sections can be improved.
Answer: The final application field was added in the abstract part as “Therefore, these newly CS/Gr modified bamboo fiber fabrics hold great promise for antibacterial application in home decoration and clothing textiles.” and the test equipment and instrumentation section was deleted, the test temperature was at 26 ℃.
What is the moisture regain of bamboo fibers and what do authors mean by completely dried? Bone-dry?
Answer: Yes, completely dried means bone-dry.
How much amount of APTES was added? What is the significance of choosing and using APTES cross-linking agent?
Answer: The mass fraction of APTES is 2 % to the CS solution. The amino group of chitosan is in the second place in the molecular structure and its performance is weak. Therefore, adding APTES crosslinking agent to strengthen amino group is easy to form composite materials closely linked with graphene.
Table 1 has listed CS% as 0.1% wheres text from 111-115 is describing as 1% of CS. Inconsistent use. What test conditions were followed for section 1.3.3 At what resolution were IR spectra capture?Author listed that graphene sheets cause physical damage, is this validated in current work and any data? How statistically significant is data reported in section 2.2?
Answer: Sorry, 1% has been changed by 0.1%. The test conditions is in general room temperature environment. The resolution of infrared spectrum is 1.5 cm-1. Graphene sheets cause physical damage also can be found in reference [11]. The average value of each data was obtained through more than three antimicrobial experiment repeats, and the difference was controlled within 3 %.
Matured cotton's longitudinal cross-sectional images are normally collapsed tube and kidney like structures and not striped textured. Needs confirmation and correction of reporting either cotton or bamboo fibers.
Answer: Ok, it`s bamboo fibers and the tubular structure replaces the stripe structure.
Why are authors referring to reference 13 when observations are evident from Figure 7. Is Figure 7 from previous work? If so, needs to clearly differentiate.
Answer: Yes, Figure 7 from our work. The reference [13] has been deleded.
Table 2 is just referring to control, C1 and C1G3, how is the trend looking for other loading ratios? And why are authors limiting sharing data for just these and not whole dataset?
Answer: The purpose is only to show that the mechanical properties of the fabrics are enhanced by adding graphene, rather than to study the changing trend of the mechanical properties of the fabrics with the amount of graphene added. And in our future work will focus on the mechanical property and compare with the difference between different Gr loadings.
Anti-microbial rate after 10 washes is reported as gradual and gentle decrease. But from 90% to 75% drop is significant. Is this decrease acceptable? And how reproducible is this data? Relative decrease after 10 washes is reported as self polymerization and contributing to increase in wash durability. This is reported as a theory and needs further validation in reference to current work.
Answer: Yes, this decrease is acceptable. According to the strandard of GB/T 20944-2008, the evaluation of the antimicrobial effect of Escherichia coli is greater than or equal to 70 %, which indicates that the sample has antimicrobial activity. The durability of fabrics has been proved by antimicrobial experiments. The antimicrobial rate can be seen in Figure 9. Text is reported as decreasing.
Also, Figure 9 is demonstrating decreasing trend whereas text is reported as increasing. Contradicting statements and observations. Needs further clarification. Needs clarification of representing L, a, b and/or L*, a*b as reported in Figure 10 What does authors mean by agglomeration phenomenon and how does it apply in the current work?
Answer: Increasing trend had been changed into decreasing. It shows that although the size of graphene powder after grinding decreases, there is still agglomeration phenomenon. The reason may be that with the increase of graphene mass fraction, the distance between graphene lamellae decreases, the probability of contact increases, and agglomeration easily occurs, which leads to the increase of graphene sheet diameter.
Only tensile properties are reported. How do other textile properties vary? Burst strength, air permeability, weight pick-up, bending rigidity, drape? These are not reported through out the manuscript.
Answer: Yes, this manuscrip focused on the antimicrobial properties, and also during the CS/Gr modification, we hope the mechanical properites having no influence or at least no decrease. For the future work, the burst strength, air permeability, weight pick-up, bending rigidity, drape and other test will be evaluated with good antimicrobial properties.
C1 has 75%, C3 has 100% of antimicrobial rates, and G1 = 94.5%, G3 = 98%. Based on these observations, should C3G3 be the best combination? Why is final loading restricted to C1G3? Conclusion section needs work on describing and summarizing the key highlights.
Answer: Chitosan is mentioned in the abstract of this article as an adhesive additive. Therefore, it is not only referring to the antimicrobial rate to select the best formula, but also approaching 100% of the antimicrobial rate of C1G3. At the same time, other properties are better and the cost is saved as much as possible.
-------------------------------------------------
We appreciate for Editor and Reviewers’ warm work earnestly, and hope that the correction will meet with approval. Once again, thank you very much for your comments and suggestions.
Yours sincerely,
Yan Wu and Bian Yuqing

Round 2
Reviewer 1 Report
The authors have satisfactory address all the problems included in the previous version, and now the work is acceptable for publication in Polymers.